# Similarity-Based Intent Detection Using an Enhanced Siamese Network

Anonymous Full Paper
Submission 12

## Abstract

In Natural Language Understanding (NLU), intent detection is crucial for improving human-computer interaction. However, traditional supervised learning models rely heavily on large annotated datasets, limiting their effectiveness in low-resource scenarios with limited labeled data. Siamese networks, which are effective at learning similarity-based representations, provide a promising alternative by enabling few-shot learning. However, Siamese networks typically rely on contrastive loss or triplet loss, both of which introduce challenges. This study introduces a similarity-based intent detection model using an enhanced Siamese network to address these limitations. Our model employs Manhattan, Euclidean, and Cosine similarity metrics combined with a fusion layer to improve intent classification accuracy. We evaluated the model on the Airline Travel Information System (ATIS) and SNIPS datasets and demonstrated its superiority over state-of-the-art methods, particularly in low-resource and few-shot learning scenarios. The results highlight significant accuracy gains while maintaining computational efficiency, making it a robust solution for real-world dialog systems.

## 1 Introduction

In today's interconnected world, dialog systems such as chatbots play a crucial role in facilitating human-machine interaction across applications such as customer service and digital assistants. Central to these systems is intent detection, which classifies users' utterances into predefined classes. A key challenge in intent detection is its reliance on labeled data, making data acquisition and annotation labor-intensive, time-consuming, and costly. Moreover, models often struggle with out-of-domain or unseen intents, reducing their real-world adaptability.

Traditional approaches to intent detection, such as rule-based methods and shallow machine learning, struggle with the complexities of natural language. The rise of deep learning models has significantly improved performance under the supervised learning paradigm. However, these approaches require large annotated datasets for each intent class, which limits their adaptability to new labels.

Siamese networks are designed to compute the similarity or dissimilarity between pairs of inputs, making them well suited for tasks where a model must identify novel classes based on minimal labeled examples [1]. These architectures have shown effectiveness in various domains, including computer vision [2], text similarity [1], and domain representation [3]. However, Siamese networks are often trained with contrastive loss [4], or triplet loss [5]. Both approaches aim to minimize the distance between similar pairs while maximizing the distance between dissimilar pairs. Triplet loss, in particular, requires the careful selection of triplets, consisting of an anchor, a positive example (similar to the anchor), and a negative example (dissimilar to the anchor). This selection process can be computationally expensive and is sensitive to the choice of margin parameter.

To address these challenges, this study proposes an enhanced intent detection model using a similarity-based Siamese network with multiple distance layers and a fusion layer. The fusion layer improves the similarity measures between sentences with the same intent and helps to better separate dissimilar intents using a binary classification approach. This eliminates the need for triplet loss, reducing model complexity while maintaining the model's ability to differentiate between similar and dissimilar intents.

Our main contributions are as follows:

- We learn sentence representations using an encoder in a Siamese network which serves as a feature extractor.

- We demonstrate the efficacy of using multiple distance layers combined with a fusion layer through ablation studies.

- In comparison to benchmark approaches, the proposed model has shown state-of-the-art (SOTA) performance, with accuracy well above 99%.

## 2 Related Work

Early intent detection relied on rule-based systems and statistical models, such as HMM, SVM, and Naïve Bayes, which struggled with scalability, semantic nuances, and context. [6] linked seen and unseen intents using manual attributes such as "action,"

"object," "location," and "time." Meanwhile, [7] and [8] improved intent detection using prosodic cues and n-grams, although extensive feature engineering was still required [9].

The introduction of deep learning revolutionized intent detection by enabling neural networks to automatically learn data patterns and representations. [10] applied a modified RNN with pre-trained embeddings for dialog act classification. CNNs, introduced by [11] for encoding, struggled with long-range dependencies, which [12] addressed through dual feature fusion with capsule networks. Similarly, [13] proposed a Bi-model based RNN for joint intent detection and slot filling, achieving state-of-the-art results on benchmark datasets.

Few-shot learning approaches have further advanced the field by allowing rapid adaptation to unseen intents with minimal labeled data. For instance, [14] leveraged pretrained models for few-shot intent detection, overcoming large-scale data dependency challenges. Building on this, [15] introduced self-supervised pretraining with prototype-aware attention for few-shot intent detection, which improved performance in scenarios with limited labeled data.

Siamese networks, widely successful in computer vision [2], have been adapted for few-shot learning in text classification. [1] applied Siamese networks to measure text similarity in ambiguous domains, while [3] extended them to learn semantic relationships with triplet loss in domain-specific contexts. Despite their success, Siamese networks with triplet loss are computationally expensive and sensitive to margin selection [16, 17].

While these approaches have significantly advanced intent detection, they often require extensive preprocessing and computational resources. Our approach builds on these advancements by integrating combined distance metrics for similarity checking, balancing efficiency and accuracy. This method bypasses large-scale triplet selection and margin tuning, making it more robust to noisy data and better at generalizing across domains.

## 3  Methodology

Figure 1 shows the architecture of the proposed similarity-based intent detection model using a Siamese Neural Network. The model consists of two identical subnetworks, a distance layer, a metric transformation layer, a fusion layer, and dense layers. The details of the model are presented in the following subsections.

### 3.1  Data Preprocessing

In the proposed model, the dataset was preprocessed by first creating pairs of texts and their corresponding intents. For each pair, a label of 1 was assigned if the intent matched the actual intent of the text, and

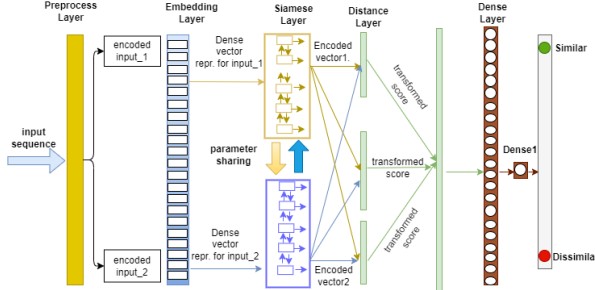

**Figure 1.** Proposed similarity based Siamese architecture

0 otherwise. Both positive pairs(matching intents) and negative pairs (non-matching intents) were used to help the model learn to distinguish between similar and dissimilar intents. The text and intent were then tokenized using the Keras Tokenizer, which converted each word into a unique integer based on its frequency in the entire dataset. This process involved fitting the tokenizer on both the text and intent columns to build a shared vocabulary.

After tokenization, each text and intent pair was transformed into numerical sequences. To ensure uniform input dimensions for the model, the sequences were padded to a fixed length. Specifically, each sequence was padded to a maximum length of 45 tokens for the ATIS dataset and 36 tokens for the SNIPS dataset for efficient batch processing.

### 3.2  Embedding Layer

An embedding layer is created to convert the input sequences into dense vector representations. The embedding weights were initialized with pretrained Word2Vec embeddings trained on 100 billion words from Google News [18]. The pretrained embeddings help capture the semantic relationship between words, where words with similar meanings have similar vector representations, aiding in better text processing. The embedding layer is shared between both input sequences to ensure identical embeddings for the same words, regardless of their position in the pair. Each word was represented by 300-dimensional vectors, with padding applied to maintain a uniform sentence length. Formally, for an utterance of length $T$, the $i^{th}$ word is mapped to $d$-dimensional embedding. Let $X_1 = [x_{1,1}, x_{1,2}, \ldots, x_{1,T}]$ be the first input sequence of length $T$ and $X_2 = [x_{2,1}, x_{2,2}, \ldots, x_{2,T}]$ be the second input sequence of length $T$, both mapped to $d$-dimensional embeddings. The embedding matrices $E_1$ and $E_2$ can be expressed as:

$$E_1 = [E(x_{1,1}), E(x_{1,2}), \ldots, E(x_{1,T})] \quad (1)$$

$$E_2 = [E(x_{2,1}), E(x_{2,2}), \ldots, E(x_{2,T})] \quad (2)$$

Where $E_1, E_2 \in \mathbb{R}^{T \times d}$.

## 3.3 Siamese Layer

The proposed model employs Bidirectional long short-term memory (BiLSTM) as the subnetwork of the Siamese layer. BiLSTMs are particularly effective for processing sequences of data [19]. Unlike traditional feedforward neural networks, BiLSTMs have connections that form directed cycles, enabling them to maintain a hidden state that captures information from the previous steps in the sequence. This makes BiLSTMs particularly effective for tasks in which the order and context of input data are crucial, such as language modeling [20], time series prediction [21], and sequence-to-sequence tasks [22].

In the proposed model, the embedded sequences $E_1$ and $E_2$ are passed through a BiLSTM, which processes the sequence step by step and generates a hidden state at each step. The recurrence relations for the hidden states at time step $t$ for the first and second sequences can be expressed as:

$$h_t^{(1)} = \sigma(W_h \cdot h_{t-1}^{(1)} + U \cdot E(x_{1,t}) + b_h) \quad (3)$$

$$h_t^{(2)} = \sigma(W_h \cdot h_{t-1}^{(2)} + U \cdot E(x_{2,t}) + b_h) \quad (4)$$

where $h_t^{(1)}$ and $h_t^{(2)}$ are the hidden states of BiLSTM at time step $t$ for the first and second sequences, respectively. $\sigma$ represents the activation function, $W_h$ and $U$ are weight matrices, and $b_h$ is the bias term. The final hidden states, $h_T^{(1)}$ and $h_T^{(2)}$, represent the encoded information for the complete sequences $X_1$ and $X_2$.

## 3.4 Distance Layer

To compute the similarity between encoded inputs $h_T^{(1)}$ and $h_T^{(2)}$, a distance layer with three different metrics: Euclidean distance, Cosine similarity, and Manhattan distance. These metrics were chosen carefully for their complementary strengths. Euclidean distance provides overall similarities, while Cosine similarity focuses on the orientation of vectors [23]. Manhattan distance captures absolute differences and is efficient for high dimensional data [24]. This combination enhances the model's ability to generalize across varied intents and domains.

$$D_{\text{Euclidean}} = \sqrt{\sum_{i=1}^{d} \left( h_T^{(1)}[i] - h_T^{(2)}[i] \right)^2} \quad (5)$$

$$D_{\text{Manhattan}} = \sum_{i=1}^{d} \left| h_T^{(1)}[i] - h_T^{(2)}[i] \right| \quad (6)$$

$$D_{\text{Cosine}} = 1 - \frac{h_T^{(1)} \cdot h_T^{(2)}}{\left\| h_T^{(1)} \right\| \left\| h_T^{(2)} \right\|} \quad (7)$$

Where $D_{\text{Euclidean}}$, $D_{\text{Manhattan}}$, and $D_{\text{Cosine}}$ represent the Euclidean, Manhattan, and cosine similarity distance metrics, respectively.

A logarithmic function was used to scale the distances for better learning and generalization [25]. The transformed distance values are then concatenated into a feature vector that captures multiple types of similarities between the two input sequences.

## 3.5 Dense Layer

The feature vector is passed through dense layers to refine the similarity score. The first dense layer was activated with ReLU function to learn complex patterns from the concatenated distance metrics. This layer ensures that the different distance metrics are jointly processed, enhancing the model's ability to capture nuanced relationships.

Another dense layer with a sigmoid activation function was added to output the probability scores.

$$\hat{y} = \sigma(z) = \frac{1}{1 + e^{-z}} \quad (8)$$

where $z$ is the output from the final layer before the activation function.

This score represents the likelihood that the input sequences corresponds to same intent.

For training, we used binary cross-entropy loss as the objective function, given that the task is framed as a binary classification problem: determining whether two input sequences correspond to the same intent. The binary cross-entropy loss is defined as:

$$\text{Loss} = -\frac{1}{N} \sum_{i=1}^{N} (y_i \log(\hat{y}_i) + (1 - y_i) \log(1 - \hat{y}_i)) \quad (9)$$

where $N$ is the total number of samples, $y_i$ is the ground truth label (0 or 1), and $\hat{y}_i$ is the predicted probability for sample $i$.

This loss function ensures that the model effectively learns to distinguish between similar and dissimilar intent pairs by minimizing the error in predicting similarity probabilities.

# 4 Experimental Study

## 4.1 Dataset

To verify the validity of our proposed model, we conducted experiments on two widely used NLU datasets, ATIS [26] and SNIPS [27], chosen for their complementary characteristics: ATIS, with 4978

training samples, 21 intents, and 128 slots [28], offers a domain-specific challenge focused on flight-related information, while SNIPS, with 15884 utterances, 7 intents, and 72 slots, provides a broader, more diverse set of challenges across various domains such as weather and entertainment, offering a comprehensive test bed for evaluating our model's effectiveness and generalizability.

## 4.2 Experimental Setup

In the proposed model, We set the BiLSTM encoders to 128 hidden units and used the ReLU activation function. To further improve training, we apply a recurrent dropout and regular dropout rate of 0.5 to randomly drop some units. The batch size was set to 32, and only ten epochs were used to train the model due to computational cost constraints. A learning rate of 0.001 was used with the Adam optimizer [29]. For the dense layers, we set the unit to 32 and 1, respectively.

For both the ATIS and SNIPS datasets, the `train_test_split` function from scikit-learn library was used to split the data into an 80% training set and a 20% test set, following the practice used in previous studies [28].To ensure robust result, the experiments were conducted with five different random seeds, and the average performance across these runs was reported. Accuracy was used as the evaluation metric, as it is the most widely adopted metric in existing models [30], [5], [31].

The model was implemented using the TensorFlow framework and trained on a machine equipped with an Intel Core i7 processor and 16.0 GB of RAM. Due to these hardware limitations, we focused on optimizing model performance within the constraints of the available computational resources.

## 4.3 Comparative Method

To further demonstrate the efficiency of the proposed model, we identified the best-performing settings of our model and subsequently compared it with the following baseline models:

- **C2A-SLU** [32]: This uses a contrastive attention mechanism to compare input sets and extract features for intent detection.

- **LIDSNet** [30]: A Siamese model with triplet loss was to reduce the distance between anchor and positive examples relative to negative examples.

- **BERT+PSN** [33]: Proposes a pseudo Siamese Network for intent detection using BERT encoders.

- **SN-TripletLoss** [5]: Proposes a Siamese network with a triplet training framework.

# 5 Results and Discussion

The performance results of the proposed Siamese network across the ATIS and SNIPS datasets, as presented in Table 1, demonstrate the significant impact of distance metric selection on intent detection task. The choice of distance metrics plays a role in determining the model's ability to generalize across datasets, particularly when the datasets vary in linguistic diversity and domain specificity. A detailed analysis of these results reveals key patterns regarding how individual and combined metrics influence model performance, providing valuable insights into the effectiveness of the Siamese network for intent detection.

The models employing individual distance metrics: `Manhattan_distance`, `Cosine_similarity`, and `Euclidean_distance`, display notable variations in performance across the two datasets. On ATIS dataset, Manhattan and Euclidean metrics achieve relatively high accuracy with score of 95.41% and 95.42%, respectively. This can be attributed to the structured and domain-specific nature of ATIS queries, which often involve repetitive patterns and similar syntax. Metrics like Manhattan and Euclidean are particularly well-suited for this scenario because they measure numerical differences and geometric distances between vector representations. These metrics help the model capture small differences in query formulations, which is essential for distinguishing intents that are syntactically close but semantically distinct, such as flight queries differing by destination or departure time.

However, the same metrics do not perform as well on the SNIPS dataset, with accuracies dropping to 86.10% (Manhattan) and 85.23% (Euclidean). SNIPS contains more diverse and varied language inputs, covering multiple domains and informal phrasing patterns. This variability makes Manhattan and Euclidean metrics less effective, as they struggle to handle the semantic richness and lexical variability present in the dataset. In this context, the reliance on numerical distance alone limits the model's ability to capture the underlying meaning of the queries.

On SNIPS, Cosine similarity performs slightly better than Manhattan and Euclidean metrics, achieving an accuracy of 85.75%. Cosine similarity focuses on the angular relationship between vectorized inputs, disregarding magnitude differences. This makes makes it more suitable for datasets like SNIPS, where different word choices and query lengths might convey the same intent. Cosine similarity helps the model recognize semantic alignment between different formulations of the same query, even when the exact phrasing differs significantly.

The result show a substantial improvement in performance when multiple metrics are combined,highlighting the benefits of metric fusion.

Model4 ( Euclidean + Cosine) achieves the highest accuracy ob both datasets, with 99.81% on ATIS and 99.67% on SNIPS. The success of this combination suggests that Euclidean distance and Cosine similarity play complementary roles in capturing intent. Euclidean distance provides a measure of positional and magnitude-based differences, which is useful for distinguishing between queries with overlapping terms but different meanings. On the other hand, Cosine similarity captures the semantic relationships between queries, allowing the model to generalize across variations in language use.

Similarly, Model6 (Cosine + Manhattan) demonstrates strong performance, especially on SNIPS, with an accuracy of 99.36%. This combination leverages Manhattan distance's ability to measure positional differences along with Cosine similarity's strength in detecting semantic alignment. The improved performance on SNIPS reflects the importance of accounting for both the semantic meaning and the positional variations in user queries, especially when dealing with multi-domain data.

However, adding a third metric does not always lead to further improvements. Model7 (Cosine + Manhattan + Euclidean), which combines all three metrics, achieves slightly lower accuracy than simpler combinations, with 99.65% on ATIS and 98.91% on SNIPS. This drop in performance can be attributed to redundancy between the metrics, as well as the increased complexity of balancing the influence of multiple metrics during training. In some cases, adding more metrics introduces noise and makes it harder for the model to learn effectively, leading to overfitting or diminishing returns in accuracy. These results highlight the importance of careful metric selection, as simpler combinations may often be more effective than using all available metrics.

The ability of the fused models to perform consistently across both datasets indicates that metric fusion improves the generalization of the Siamese network. While individual metrics perform well only on specific datasets, such as Manhattan and Euclidean on ATIS or Cosine similarity on SNIPS, their combination allows the model to leverage the strengths of each metric. This enables the model to handle both domain-specific queries (ATIS) and multi-domain queries (SNIPS) effectively. For instance, the combination of Euclidean and Cosine metrics captures both the magnitude-based distinctions needed for structured queries and the semantic alignment needed for diverse inputs.

The logarithmic transformation applied to each metric further enhances the model's generalization. By normalizing the values of the metrics, the transformation smooths out large differences and prevents any single metric from dominating the fusion layer. This ensures that the model benefits equally from the complementary strengths of multiple metrics, leading to improved convergence during training.

The results suggest that metric selection should align with the nature of the dataset. For datasets like ATIS, where queries are relatively uniform and domain-specific, metrics that measure numerical differences or geometric distances are more effective. In contrast, for datasets like SNIPS, where semantic richness and diversity are key characteristics, Cosine similarity or combinations of metrics that capture both semantic and positional differences yield better performance.

Additionally, the performance decline observed in Model7 highlights the need to balance model complexity with performance gains. While combining metrics can enhance generalization, using too many metrics may lead to redundancy and hinder performance. These findings emphasize the importance of selecting complementary metrics that align with the specific requirements of the task and dataset.

**Table 1.** Performance of Different Versions of the Proposed Model

| Models | Distance Metrics | Accuracy (%) | |
| --- | --- | --- | --- |
| | | **ATIS** | **SNIPS** |
| Model1 | Manhattan | 95.41 | 86.10 |
| Model2 | Cosine Similarity | 95.32 | 85.75 |
| Model3 | Euclidean | 95.42 | 85.23 |
| Model4 | Euclidean + Cosine | **99.81** | **99.67** |
| Model5 | Euclidean + Manhattan | 99.80 | 86.13 |
| Model6 | Cosine + Manhattan | 99.74 | 99.36 |
| Model7 | Cosine + Manhattan + Euclidean | 99.65 | 98.91 |

# 6 Comparison with State-of-the-Art Models

To evaluate our proposed model, we compared the best performing setting, Model4 against state-of-the-art models. The results in Table 2 demonstrate the superiority of our model on both ATIS and SNIPS datasets.

**Table 2.** Comparison with Published Results on ATIS and SNIPS Datasets

| Model | ATIS (%) | SNIPS (%) |
| --- | --- | --- |
| C2A-SLU [32] | 96.84 | - |
| LIDSNet [30] | 95.97 | 98.00 |
| BERT+PSN [33] | - | 92.89 |
| SN-TripletLoss [5] | 99.56 | 99.31 |
| **Ours** | **99.81** | **99.67** |

Our model outperformed the C2A-SLU model by 3.06% on the ATIS dataset. This improvement can be attributed to the fact that contrastive learning primarily focuses on representation learning, which

may not be as directly optimized for task-specific objectives as in the proposed approach.

Compared to LIDSNet, which uses triplet loss for training, our model achieved a 4% higher accuracy on the ATIS dataset and a 1.70% improvement on the SNIPS dataset. Compared with SN-TripletLoss, which also uses a triplet loss framework, our model showed an improvement in accuracy of 0.25% and 0.36% on the ATIS and SNIPS datasets, respectively. The more efficient performance of our model can be attributed to its reliance on distance metrics, which involve fewer hyperparameters and are less prone to the challenges of tuning margin parameters, learning rates, and triplet mining strategies that often lead to suboptimal performance in LIDSNet and SN-TripletLoss.

In contrast to BERT+PSN, which uses a pseudo-Siamese network for few-shot intent detection, our model demonstrated a notable 7.3% improvement in accuracy on the SNIPS dataset. This significant margin underscores the robustness of our approach, particularly in scenarios with limited labeled data for intent detection.

## 7 Conclusion

This study presented a novel intent detection approach using an enhanced Siamese network that integrates multiple distance metrics with a fusion layer. The proposed model demonstrated superior performance on the ATIS and SNIPS datasets, outperforming state-of-the-art methods. The combination of Manhattan, Euclidean, and Cosine similarity metrics proved crucial in handling diverse and domain-specific tasks, improving generalization and reducing dependence on annotated datasets. By simplifying the architecture and minimizing hyperparameter tuning, our model offers an efficient, scalable solution, particularly in low-resource environments.

Despite the promising results, this study has some limitations. A key limitation lies in the use of ATIS and SNIPS datasets, which, although widely adopted benchmarks, have become overused in recent research. As a result, the performance gains observed on these datasets may not translate directly to real-world applications with more complex and evolving intent structures. Additionally, while the fusion of multiple metrics improved accuracy, the individual metrics produced only marginal improvements. This suggests that the impact of individual metrics might be limited when dealing with datasets that are not as saturated with patterns as ATIS. Another limitation is the absence of a detailed investigation into the learned representations and the specific contributions of each metric. This leaves room for further exploration into how the metrics complement each other.

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
