# OpenReview forum: "Similarity-Based Intent Detection Using an Enhanced Siamese Network"
_NLDL.org/2025/Conference — Submitted to NLDL 2025_

### Official Review · Reviewer_8o6U · 2024-09-18
**Unverified claims, poor methodology, and absent analysis for overused dataset of Intent Detection**

**Confidence:** 4

**Summary:**

This work introduces a study on intent classification through the lens of siamese networks. It uses 2 datasets for intent (ATIS and SNIPS) , trains a BiLSTM in a siamese setting and tests a variety of similarity metrics.

**Strengths:**

The setting is relevant

**Weaknesses:**

- The paper is full of claims and scenarios that arent verified. This already starts in the abstract, e.g. line 15 talking about low-resource or zero-shot learning scenario's. None of this is actually debated later in the paper. The method is never verified in a zero-shot learning setting, and the computational complexity (or even hardware for that matter) is never discussed.
	- Line 67-68: "This approach brings sentence representations of the same intent closer to the latent space, thereby enabling effective intent detection.". This is never specifically shown.
	- Line 193, 194, 195: This doesnt contain an actual specific reason for why they were selected.
	- The large majority of plots are uninteresting the scaling of axis is consistently bad. Figure (c) reports on a difference in the 4th decimal place for accuracy, where the entire plot is a flat line. This is not a unique occurence for the graphs.
	- It is unclear to me how the downstream predictions are actually done. The distance layer measures the distance using the metrics and when multiple are used they are concatenated in the fusion layer?. So then 1 or 2 metrics are used an input for the dense layer? The whole concept is vaguely explained. In the end, figure 1 and the architecture predicts a similarity score over 2 inputs. But nowhere is explained how this then goes to the predictions. The result is also a single dense node, which is also the similarity score? This also results in metrics being learned twice? One for the metric layer and one in the end?
	- The https://paperswithcode.com/sota/intent-detection-on-atis mentions many works that are actually much closer to the scores reported, but only matches with 1 of the baselines (LIDSNet). What worries me more is the already high performances on this task. Especially the ATIS dataset is an already milked out benchmark in terms of performance. The work also shows that performance is high from epoch 0, with sometimes differences in the 3rd decimal after 10 epochs.
	- In similar fashion, this work would take number 1 position in this benchmark but this could be due to stochastic variations
	- The results section and analysis is essentially absent. There is nothing but accuracy in table form or graph form.
	- There is no investigation done about any of the representations that are learned or how well the introduced layers work. What are the effects of a distance layer? It seems like the "metric transformation layer" is nothing but a logarithmic transformation? Is this valid here? No arguments are provided for these crucial choices other than "other works do it", despite those being different settings.
	- The dataset is also poorly described in general. The original dataset also already comes with a training (4978) and test split (893), so why the additional train_test_split was done is unclear.
The overall story is just incoherent, and the related work and literature search feels like searching "intent classification" on Google Scholar and stringing the sources together. There is not a single thing being specifically addressed in the end. Even the comparisons to mentioned base-lines are largely uninteresting for the same mentioned reason before: ATIS and SNIPS seem to be milked out benchmarks where the measurement over their performance alone has become largely uninteresting

**Justification:**

This work explores siamese networks for intent classification, specifically by using intermediate layers and metrics for learning the final similarity "measure". They largely accomplish this by introducing extra layers before learning the similarity metric (a "distance layer", "metric transformation layer", "fusion layer" and a resulting dense layer). The validity of introducing these layers and what occurs in them is not discussed. The work is very vague in terms of what it tries to show and/or accomplish. Claims are made are largely unsubstantiated (e.g. it working in a zero-shot setting or low resource). The evaluation is nothing but the same 12 uninspiring and uninforming plots and 2 tables comparing to previous results. The final result is a paper that feels like the result of a bachelors project, rather than actual contributing research.

---

> ### Author Rebuttal · Authors · 2024-10-23
>
> **1. The paper is full of claims and scenarios that arent verified. This already starts in the abstract, e.g. line 15 talking about low-resource or zero-shot learning scenario's. None of this is actually debated later in the paper. The method is never verified in a zero-shot learning setting, and the computational complexity (or even hardware for that matter) is never discussed.**
>
> We appreciate this observation and agree that the mention of zero-shot learning was premature. We intended to demonstrate that our approach shows potential in low resource scenarios, but the reference to zero-shot was misleading. We have revised the abstract and introduction to reflect the intended low-resource scenario, ensuring that the claims are consistent with the results.
>
> **2. Line 67-68: "This approach brings sentence representations of the same intent closer to the latent space, thereby enabling effective intent detection.". This is never specifically shown.**
> We appreciate this insightful comment. While our current work focuses primarily on the performance outcomes of the fused distance metrics, we acknowledge that a more detailed analysis of how sentence representations are projected into the latent space is needed. This is an important aspect, as it directly impacts the model's ability to distinguish between similar and dissimilar intents. In future work, we plan to conduct a deeper investigation into the latent space representations, including techniques such as visualizing embeddings with t-SNE or PCA to better understand the clustering behavior. In the revised manuscript, we have acknowledged this limitation and position it as an area for further research.  line 521-526
>
> **3. Line 193, 194, 195: This doesnt contain an actual specific reason for why they were selected.**
> We have expanded the methodology section to explicitly justify these choices with supporting references. Line 216-226
>
> **4. It is unclear to me how the downstream predictions are actually done. **
>
> We have revised the methodology to reflect how the prediction is  done. line 141-267
>
> **ATIS dataset is an already milked out benchmark in terms of performance.**
>
> We acknowledge that the ATIS dataset is a well-established and widely used benchmark for intent detection and has been studied extensively by the research community. However, ATIS still serves as an essential benchmark for evaluating new architectures, especially in intent detection and slot filling tasks models. By using ATIS, we aim to provide comparability with prior works and ensure that the proposed architecture meets established standards in intent detection performance.
>
> **5. Justification for Using Existing Benchmarks**
> Despite the saturation of performance on ATIS, it remains relevant as it allows for fair and direct comparisons with previous baselines, including LIDSNet and other high-performing models mentioned in Papers with Code. While many existing models achieve high scores, these results are not always obtained using Siamese networks with multiple distance metrics, which is the primary contribution of our work. The novelty lies in how our model fuses multiple metrics to enhance generalization, rather than solely improving accuracy.
>
> **6. Clarifying Baseline Comparison**
> We compared our results with LIDSNet because it closely aligns with our task setup and architecture. Other high-performing models mentioned in the benchmark often rely on transformer-based architectures or extensive pre-training on large datasets, which are beyond the scope of this work. Our goal is to explore the potential of similarity-based architectures for intent detection with fewer parameters, and we aim to build upon these results in future research with more diverse datasets.
>
> **Addressing Early High Performance and Small Improvements Over Epochs**
> We recognize that the high initial performance and small incremental gains over time reflect the saturation of the ATIS benchmark. This is a common phenomenon for tasks like intent detection on extensively studied datasets, where models quickly learn predictable patterns.
> However, our analysis highlights a key difference:
> •	When using individual distance metrics (e.g., Euclidean, Manhattan, or Cosine similarity) in isolation, the performance stabilizes early, with minimal gains observed over subsequent epochs.
> •	In contrast, when multiple metrics are combined through fusion, the model achieves significantly improved performance, demonstrating the value of metric fusion in capturing complementary information. This fusion approach enables the model to refine predictions over time and achieve better generalization across intents.
>
> Even seemingly small improvements (e.g., changes in the third decimal place) reflect the effectiveness of metric fusion in making more precise predictions, which individual metrics alone might miss. This shows that the combined approach offers a better trade-off between multiple types of similarities, leading to more meaningful performance gains

---

### Official Review · Reviewer_Mdjj · 2024-09-20
**Review for "Similarity-Based Intent Detection Using an Enhanced Siamese Network"**

**Confidence:** 3

**Summary:**

The paper proposes a novel variant of Siamese networks for intent detection in which multiple notions of distance (esp. Manhattan, Euclidean, Cosine) are merged to achieve improved results. Experiments on two data sets revealed that combining Euclidean and Cosine distances yields the best results.

**Strengths:**

The strengths of the paper are:
- Siamese Networks are a quite popular and flexible architecture, such that progress in this architecture has large potential impact in the community.
- The proposed architectural change is quite straightforward and should be simple to implement in many domains.
- The experimental results appear to show a clear benefit of the proposed change.

**Weaknesses:**

The weaknesses are:
- Perhaps most importantly, I think the paper overclaims its contribution. The notion of distance layers and alternative loss functions has already been discussed extensively in the Siamese Network community. Refer, e.g., to Chicco (2021): https://doi.org/10.1007/978-1-0716-0826-5_3 or to Wang and Liu (2021): https://openaccess.thecvf.com/content/CVPR2021/papers/Wang_Understanding_the_Behaviour_of_Contrastive_Loss_CVPR_2021_paper.pdf
- By contrast, the core contribution seems to be (to me) the evaluation and fusion of multiple distances, which should be made much more clear.
- Comparing the proposed model to published results in Table 2 may be misleading if different train/test splits were used. This should be specified. Ideally, at least one strong baseline should be trained and tested on the same split.
- The loss function used for training has not been specified. I assume that a contrastive loss has been used, but this should be clarified.
- It might have been instructive to analyze multiple possible embedding architectures, perhaps also using a BERT-based one.
- I am not sure that the many loss curves in Figure 2 contribute sufficiently to the paper to warrant taking that much space.

**Final Rebuttal Confidence:**

3

**Final Rebuttal Justification:**

I am not convinced that evaluating a BERT model would impose excessive computational load (by nowadays standards, this counts as a rather small model). Nonetheless, most of my points have been addressed so that I remain at my evaluation.

**Justification:**

Overall, I think the core innovation - fusing multiple distances - is substantial enough to warrant acceptance, although I would appreciate more clarity.

---

> ### Author Rebuttal · Authors · 2024-10-23
>
> **1. Perhaps most importantly, I think the paper overclaims its contribution. The notion of distance layers and alternative loss functions has already been discussed extensively in the Siamese Network community. Refer, e.g., to Chicco (2021): https://doi.org/10.1007/978-1-0716-0826-5_3 or to Wang and Liu (2021):**
>
> We acknowledged that a lot of works have discussed on Siamese Network. In this study we tried to contribute our own quota in the context of NLP(intent detection).
>
> **2. By contrast, the core contribution seems to be (to me) the evaluation and fusion of multiple distances, which should be made much more clear**
>
> We have reviewed our paper to make our contribution more clearer at different sections of the paper line 012-017, 063-073, 126 -132, 134-267. 326-460.
>
> **3. The loss function used for training has not been specified. I assume that a contrastive loss has been used, but this should be clarified.**
>
> Thank you for pointing this out. We used binary cross-entropy loss for the training, as our task involved similarity based intent classification. line 063-070, 254-267.
>
> **4. It might have been instructive to analyze multiple possible embedding architectures, perhaps also using a BERT-based one.**
> We recognize the value of exploring different embedding architectures. However, due to computational constraints, we focused on the BiLSTM-based approach. In future work, we plan to explore BERT-based embeddings to assess the generalizability of our method.
>
> **5. I am not sure that the many loss curves in Figure 2 contribute sufficiently to the paper to warrant taking that much space.**
>
> Thank you for your feedback. We acknowledge that some of the graphs may appear redundant, and we have removed them

---

### Official Review · Reviewer_6Brq · 2024-09-25
**The claim in the conclusion is not well-founded**

**Confidence:** 4

**Summary:**

This work proposes a modification to the contrastive loss, where an extra dense layer is added after the distance calculation. This allows for using several different distance measures and combine the result though the dense layer to a single "similarity score".

The authors claim that this kind of model beats state-of-the-art models in low-resource and zero-shot scenarios.

**Strengths:**

The new thing in this work is the added dense layer after measuring the distances, and the fact that this enables using multiple different distance measures in the model.

Questions:


Q1: In 3.4, it says "A logarithmic function was used to normalize the distances for better learning and generalization."
Do you just mean you took the logarithm of the distance? Or did you use the logarithm to do some kind of normalization?


Q2: In the intro, it says: "The fusion layer improves the similarity measures between sentences with the same intent, eliminating the need for triplet selection"
Does this mean that you do not use negative pairs in your training?
And if that is the case, where then is the model's incentive to keep dissimilar things apart?
Should it not just place everything together in that case?


Q3: Since you mention reducing model complexity, did you compare with a model using standard contrastive loss, so without the extra dense layer? If so, please be explicit about the change in performance and size.

**Weaknesses:**

**W1: Is the trained models for this work actually zero-shot?**

In the abstract and introduction it is mentioned several times that this work is a zero-shot approach.
However in section 4.2, there is no mention of any intents being left out of the training data.
The compared baseline model from [31] is few-shot and the compared baseline model from [32] is zero-shot,
so if the trained models from the paper are not zero-shot different baseline models should be chosen.
So, it is unclear whether the paper is actually doing what it claims to do.


**W2: Generalizability**

There is no mention in the paper of training multiple seeds or averaging over a number of runs.
If only 1 model was trained for each combination of distance metrics, no conclusions should be drawn about the generalizability of the results.
In particular, it is not possible to conclude from these results that the proposed model will in general achieve better accuracy than all the models used as baselines.
Thus the conclusion of the paper is not well-founded.


**W3: Reproducibility**

More details are needed (could be put in an appendix) to reproduce the experiments.
In 4.2, it says that the train_test_split_function was used, but not from which library.
Used random seeds should be reported.
The paper does not mention whether the code will be released.


**W4: Related work**

Hadsell et al. "Dimensionality Reduction by Learning an Invariant Mapping" should be cited, since this work builds on contrastive loss.


[31] Xia 2021 "Pseudo Siamese Network for Few-shot Intent Generation"

[32] Xue 2021 "Intent-enhanced attentive Bert capsule network for zero-shot intention detection"

**Final Rebuttal Confidence:**

5

**Final Rebuttal Justification:**

I recommend to reject this paper.

The authors admitted that their work is not zero or few-shot, but they did not remove all the places in the article where they imply that it is few-shot (see list in my comments). They also did not remove all comparisons with few-shot work.

On top of this, there are many claims in this article which are not supported by results (see my comments).

If an article cannot be clear about what the proposed method does and does not support its claims with results, it is not ready to be published.

**Justification:**

In its current form I do not recommend this paper to be accepted.

It is unclear whether the work is actually zero-shot as it claims to be, and even if it turns out to be zero-shot, the conclusions drawn are not well-founded.

---

> ### Author Rebuttal · Authors · 2024-10-23
>
> **1. In 3.4, it says "A logarithmic function was used to normalize the distances for better learning and generalization." Do you just mean you took the logarithm of the distance? Or did you use the logarithm to do some kind of normalization?**
>
> Thanks for your observation it was an oversight, the logarithm helps in scaling distances for better model learning, not formal normalization. line 233-234
>
> **2. In the intro, it says: "The fusion layer improves the similarity measures between sentences with the same intent, eliminating the need for triplet selection" Does this mean that you do not use negative pairs in your training? And if that is the case, where then is the model's incentive to keep dissimilar things apart? Should it not just place everything together in that case?**
>
> Negative pairs are indeed used during training. For each text, pairs are formed with both matching intents (positive pairs) and non-matching intents (negative pairs). The labels 1 (for matching) and 0 (for non-matching) help the model learn to distinguish between similar and dissimilar pairs. The model's incentive to keep dissimilar pairs apart is driven by the binary cross-entropy loss. This loss function encourages the model to assign a high similarity score to pairs with the same intent (label 1) and a low similarity score to pairs with different intents (label 0).  line 142-154
>
> **3. Since you mention reducing model complexity, did you compare with a model using standard contrastive loss, so without the extra dense layer? If so, please be explicit about the change in performance and size.
>
> In our proposed model, we did not directly compare it to a model using standard contrastive loss. The focus of this study was to improve performance through the combination of multiple distance metrics and the fusion layer. The dense layer plays a role in enhancing feature extraction by combining these transformed metrics. However, future work could include an explicit comparison between models with and without the dense layer to examine the change in performance and model size.
>
> **4. There is no mention in the paper of training multiple seeds or averaging over a number of runs. If only 1 model was trained for each combination of distance metrics, no conclusions should be drawn about the generalizability of the results. In particular, it is not possible to conclude from these results that the proposed model will in general achieve better accuracy than all the models used as baselines. Thus the conclusion of the paper is not well-founded.**
>
> We have clearly stated the number of times we run the experiment line 297-300
>
> **5. Reproducibility**
>
> **More details are needed (could be put in an appendix) to reproduce the experiments. In 4.2, it says that the train_test_split_function was used, but not from which library. Used random seeds should be reported. The paper does not mention whether the code will be released.**
>
> We have now clearly stated the library used for the split at line 294 and ready to release the code under request.
>
> **Related work**
> **Hadsell et al. "Dimensionality Reduction by Learning an Invariant Mapping" should be cited, since this work builds on contrastive loss.**
> we acknowledge your suggestion and we have done that line 054.
>
> **It is unclear whether the work is actually zero-shot as it claims to be, and even if it turns out to be zero-shot, the conclusions drawn are not well-founded.**
>
> It is not a zero-shot and we have clearly stated some of the limitations of our work.

---

### Official Review · Reviewer_zcwh · 2024-10-02
**Saturating Intent-detection benchmarks**

**Confidence:** 4

**Summary:**

The paper proposes siamese bi-lstm networks combined with both euclidean and cosine similarity to solve intent detection. The suggested model effectively saturates both the ATIS and SNIPS datasets, setting a new SOTA.

**Strengths:**

The model performs very well on the described narrow task of intent detection. The paper is easy to read in general. The used architecture is quite simple, yet powerful.

**Weaknesses:**

Most papers combine intent detection with slot filling, this one focuses on this very narrow task and on only two (the most common) datasets for it.

More detailed feedback:
- I am surprised that table1 does not include an "all three" option
- Is it really common to not use train/val/test but only train/test in intent detection?
- The left part of figure1 is unclear - this is absolutely standard, why not just display the content on the right of the figure?
- Related work should include that GPT4 reaches around 90% and is thus clearly worse compared to specialized approaches. Furthermore, I would expect the current winners at paperswithcode to be cited here (and hope that you will upload your results there as well). In general, related work feels outdated with the most current paper dating back to 2021, has there really been no new stuff happening since then?
- "computational constraints" were mentioned but without going into details - how slow is training on which kind of hardware?
- tokenization is unclear, what exactly are you doing here?
- it is unclear why one should perform padding when using an LSTM.
- 202: log(dist) is NOT normalization!!! (even though it keeps the numbers low)

Some Latex remarks:
- The correct way of using ``quotation marks'' uses backquotes in the beginning and straight quotes in the end.
- please use booktabs
- its \mathbb R for the reals
- commonly the nonlinearity is called \sigma and not \phi

And some spelling:
- 049: where _a_ model
- 053: comma missing
- 056: an anchor ... a neg example
- 068: _in_ the latent space
- 072: no comma
- 075: _a_ fusion layer
- 077: comparison (without s!), then _has_ good performance (should be further specified that the performance is SOTA and well-above 99%)
- 116: generalize-> generalizing
- 186: for the _complete_ sequences

**Justification:**

Very simple architecture that clearly outperforms competitors.

---

> ### Author Rebuttal · Authors · 2024-10-23
>
> 1. I am surprised that table1 does not include an "all three" option
>
> Thanks for your observation, we have now included that in our Table 1
>
> 2. Is it really common to not use train/val/test but only train/test in intent detection?
>
> Many studies used train_test_split of 80% train and 20% test as reported in a paper titled: “A survey of joint intent detection and slot filling models in natural language understanding” and we have cited that in our paper line 297
>
> 3. The left part of figure1 is unclear - this is absolutely standard, why not just display the content on the right of the figure?
>
> We have simplified the Fig. 1
>
> 4. Related work should include that GPT4 reaches around 90% and is thus clearly worse compared to specialized approaches. Furthermore, I would expect the current winners at paperswithcode to be cited here (and hope that you will upload your results there as well). In general, related work feels outdated with the most current paper dating back to 2021, has there really been no new stuff happening since then?
>
> Thanks for your suggestion, we have included the suggested works and some recent papers published in 2024. ref 14,15, 17
>
> 5. "computational constraints" were mentioned but without going into details - how slow is training on which kind of hardware?
> It was trained on an Intel Core i7 processor, 16.0 GB RAM. On this hardware, the training time for 10 epochs was approximately 15 minutes per dataset. line 303-308
>
> 6. tokenization is unclear, what exactly are you doing here?
> Thank you for your insightful feedback. We acknowledge that our original submission did not provide sufficient detail on the tokenization process. We have now revised the Data Preprocessing sections line 142 -154
>
> In our model, we used tokenization at preprocessing stage to convert the input text and intent labels to an integer sequences using the Keras Tokenizer. This allows the model to handle natural language data efficiently. Specifically, each word is mapped to a unique integer index based on its frequency in the dataset, ensuring consistency between user queries and intent labels.
>
> 7. it is unclear why one should perform padding when using an LSTM.
> Although LSTMs are theoretically capable of handling variable-length sequences. However, when batch processing is to be used, it requires inputs to be of the same length within a batch. line 155-161
>
> 8. log(dist) is NOT normalization!!! (even though it keeps the numbers low)
> Thank you for this great observation and we have taken note and effected the correction.    line 233-234
>
> 9. Some Latex remarks:
> We have taken note, and corrected accordingly.
>
> The correct way of using ``quotation marks'' uses backquotes in the beginning and straight quotes in the end.     effected at  line 090-091.
> please use booktabs         done at line 366 (Table 1) and line 372 (Table 2 )
>
> its \mathbb R for the reals    done at line 186
>
> commonly the nonlinearity is called \sigma and not \phi       corrected  at line 199-200 and line 242.
>
> **some spelling**
> Now corrected
> line 048: where a model
> line 054: comma missing
> line 058 : an anchor ... a neg example
> line 076: no comma
> line 079: a fusion layer
> line 083: comparison (without s!), then has good performance (should be further specified that the performance is SOTA and well-above 99%)
> line 132: generalize-> generalizing
> line 213: for the complete sequences

---

### Meta-Review · Area_Chair_hCjo · 2024-11-04

**Recommendation:** Reject
**Confidence:** 3

**Metareview:**

The paper introduces a similarity based Siamese network for intent detection. Several similarities/dissimilarities between latent representations of input text and an intent are computed and combined into one representation, which is further processed for classification. The model outperforms the SOTA on two widely used datasets.

The paper is a borderline case, with somewhat divergent opinions from the reviewers. The main issues have been w.r.t. claims unsupported by results and lacking experimental design. While some concerns were addressed during the rebuttal phase, the majority agrees that the work is not ready for publication in its current state.

**Suggested Changes To The Recommendation:**

2: I'm certain of the recommendation.  It should not be changed

---

### Decision · Program_Chairs · 2024-11-06

Reject